# Intermetallic Phases Identification and Diffusion Simulation in Twin-Roll Cast Al-Fe Clad Sheet

**DOI:** 10.3390/ma14247771

**Published:** 2021-12-16

**Authors:** Barbora Křivská, Michaela Šlapáková, Jozef Veselý, Martin Kihoulou, Klaudia Fekete, Peter Minárik, Rostislav Králík, Olexandr Grydin, Mykhailo Stolbchenko, Mirko Schaper

**Affiliations:** 1Faculty of Mathematics and Physics, Charles University, Ke Karlovu 5, 121 16 Prague, Czech Republic; barbora.krivska@mff.cuni.cz (B.K.); jozef.vesely@mff.cuni.cz (J.V.); martin.kihoulou@mff.cuni.cz (M.K.); klaudia.fekete@mff.cuni.cz (K.F.); peter.minarik@mff.cuni.cz (P.M.); rostislav.kralik@mff.cuni.cz (R.K.); 2Chair of Materials Science, Paderborn University, Warburger Str. 100, 33098 Paderborn, Germany; grydin@lwk.upb.de (O.G.); stolbchenko@lwk.upb.de (M.S.); schaper@lwk.upb.de (M.S.)

**Keywords:** TEM SAED, diffusion, Al_5_Fe_2_, Al_13_Fe_4_, Boltzmann–Matano method

## Abstract

Aluminium steel clad materials have high potential for industrial applications. Their mechanical properties are governed by an intermetallic layer, which forms upon heat treatment at the Al-Fe interface. Transmission electron microscopy was employed to identify the phases present at the interface by selective area electron diffraction and energy dispersive spectroscopy. Three phases were identified: orthorhombic Al_5_Fe_2_, monoclinic Al_13_Fe_4_ and cubic Al_19_Fe_4_MnSi_2_. An effective interdiffusion coefficient dependent on concentration was determined according to the Boltzmann–Matano method. The highest value of the interdiffusion coefficient was reached at the composition of the intermetallic phases. Afterwards, the process of diffusion considering the evaluated interdiffusion coefficient was simulated using the finite element method. Results of the simulations revealed that growth of the intermetallic phases proceeds preferentially in the direction of aluminium.

## 1. Introduction

### 1.1. Aluminium-Steel Interface

Clad materials of two or more constituent metals can exhibit superior thermal and mechanical properties when compared to single metals. Aluminium-steel clad sheets combine the low density of aluminium and high strength of steel which makes them a good candidate for application in automotive, aircraft and food industries.

Nowadays, many methods have been carried out to fabricate clad sheets, such as welding, extrusion, roll bonding or cold rolling [1]. Twin-roll casting is a novel method which is of great interest for industrial applications due to a shorter production chain in comparison with conventional sheet bonding technologies. As a consequence of the high temperature and compressive stresses in the twin-roll caster, the clad strip is produced directly from the aluminium melt on a steel substrate in a single technological operation. By exclusion of the intermediate heating and substrate preparation operations, the energy and material consumption, as well as detrimental impurities, can be significantly reduced [2].

The final properties of the clad material are governed by the presence, thickness and constitution of intermetallic phases (IMC) that form at the aluminium-steel interface upon a thermal treatment [3]. A limiting thickness of the IMC for a deterioration of mechanical properties of the clad sheets was reported to be 8–10 μm [1,4]. Hence, the phases that form at the Al-Fe interface have been studied carefully in recent years [5,6,7,8].

Several kinds of crystal structure are known to form on the Al-rich side of the Al-Fe alloy system [9]. The two most commonly observed phases at the Al-Fe interface are the η-phase Al${}_{5}$Fe${}_{2}$ and θ-phase Al${}_{13}$Fe${}_{4}$ (also referred to as Al${}_{3}$Fe), respectively, Al${}_{5}$(Fe,Cr,Ni)${}_{2}$ and Al${}_{13}$(Fe,Cr,Ni)${}_{4}$ considering stainless steel [10]. The orthorhombic Al${}_{5}$Fe${}_{2}$ phase is stable for 70–73 at.% Al, the monoclinic Al${}_{13}$Fe${}_{4}$ for 74.5–76.5 at.% Al [11]. Brittleness of the IMC increases with increasing Al content and η and θ-phases were reported to have a low fracture toughness, which is undesirable for engineering applications [12].

Concerning the growth kinetics of the Al${}_{5}$Fe${}_{2}$ and Al${}_{13}$Fe${}_{4}$ phases, Sapanathan et al. [11] stated that ex situ analyses are not capable to capture nucleation and early stages of IMC growth due to their extremely fast kinetics. They have employed high temperature in situ scanning electron microscopy and showed that the θ-phase Al${}_{13}$Fe${}_{4}$, which prevailed in their case, nucleates first prior to diffusion-controlled growth of the η-phase Al${}_{5}$Fe${}_{2}$. The growth of the phases was rather towards aluminium. On the other hand, Zhang et al. [13] studied kinetics of the IMC growth at the interface of solid steel and liquid aluminium. Al${}_{5}$Fe${}_{2}$ formed first at the brazing interface towards the Al side. Its preferred crystal orientation provides a path for Al atoms to migrate through the IMC layer for further growth of the Al${}_{5}$Fe${}_{2}$ layer towards the steel substrate. Subsequently, Al${}_{13}$Fe${}_{4}$ formed at the Al${}_{5}$Fe${}_{2}$–aluminium interface during solidification and grew into the Al. The Al-Fe intermetallics finally become a barrier for further migration/diffusion of Fe and Al atoms and play an important role in determining the thickness of the IMC layer. Yang et al. [1] calculated through the lowest Gibbs free energy that the Al${}_{5}$Fe${}_{2}$ phase is the first phase to form. According to their experimental observations, firstly, the interdiffusion of Al and Fe appears at the interface. Secondly, the Al${}_{5}$Fe${}_{2}$ phase forms at the interface zone with the accumulation of Fe atoms in aluminium as the diffusion activation energy of Fe atoms in solid Al is lower than vice versa. Finally, the formed Al${}_{5}$Fe${}_{2}$ reacts with Al atoms and forms Al${}_{13}$Fe${}_{4}$ intermetallics and, simultaneously, Al atoms cross the intermetallic layer to react with Fe atoms to form an additional Al${}_{5}$Fe${}_{2}$ layer. Liu et al. [10] investigated a sample where the IMC layer between Al and steel was not continuous; hence, the position of the original Al-steel interface was recognisable. In their case, the IMC expanded to both the Al and steel side, with growth towards the aluminium side prevailing.

Grydin et al. [2] focused on the influence of the casting parameters on the intermetallics layer thickness after casting. The temperature of the steel substrate during twin-roll casting was shown to have a significant role on the mechanical properties of the clad material—increasing the steel temperature, the thickness of the IMC increased due to enhanced diffusion. The highest bonding strength was achieved with substrate preheating to 280 ${}^{\circ }$C; a further increase of the temperature led to a decrease of the bonding strength. The optimal thickness of the IMC layer was hence assessed to be ∼3 μm.

### 1.2. Diffusion

A classical description of diffusion processes is given by Fick’s second law (diffusion equation) [14], originating from a direct proportion between a mass flux and the gradient of concentration *c* together with the principle of mass conservation and the divergence theorem [15,16]. In 1D, the formula is given by
(1)$$\frac{\partial c(x,t)}{\partial t}=\frac{\partial }{\partial x}\left[D\left(c\right)\frac{\partial c}{\partial x}\right],$$
where *x* is the coordinate, *t* is the time and *D* is the diffusion coefficient.

In heterogeneous A–B binary systems, where atoms A and B are of a comparable radius, the main diffusion mechanism involves lattice vacancies. The vacancy mechanism enables the diffusing species to have different diffusion coefficients, as some atoms could easier/faster occupy the vacant lattice sites. In the large-scale diffusion, there will be a flux of mass (atoms) in one direction and a flux of vacancies in the other one. This phenomenon is known as the Kirkendall effect [17].

Different diffusivities could be represented by concentration-dependent diffusion coefficient D(c), which introduces non-linearity to Equation (Equation 1). Concentration-dependent diffusion coefficient can be determined using the Boltzmann–Matano method, based on inversion of the Fick’s second law. This method gives a formula for the diffusion coefficient D(c) based on a certain concentration profile c(x) resulting from a diffusion of a time duration *t*
(2)$$D\left(c^{*}\right)=-\frac{1}{2t}\frac{\int_{c_{L}}^{c^{*}}\left(x-x_{M}\right)dc}{\frac{dc}{dx}{|}_{c^{*}}},$$
where $c^{*}$ signs a selected concentration and $x_{M}$ is a position of the so-called Matano plane [18,19], representing an equal integral for the concentration profile on either side of the plane. The position of this plane results from a conservation condition stemming from the fact that the gain of the diffusing species found on the left-hand side must have entered by diffusion from the right-hand side (see Figure 1)
(3)$$\int_{-\infty }^{x_{M}}\left[c\left(x\right)-c_{L}\right]dx=\int_{x_{M}}^{\infty }\left[c_{R}-c\left(x\right)\right]dx,$$
where $c_{L}$ is the concentration of the studied element on the left and $c_{R}$ is its concentration on the right.

The aim of the present study is to identify the phases that form in a twin-roll cast Al-steel clad sheet and evaluate the preferential direction of diffusion at the Al-Fe interface by simulation using the Boltzmann–Matano method.

## 2. Materials and Methods

The aluminium-steel clad sheet was twin-roll cast under laboratory conditions at Padeborn University. A melt of a technical pure aluminium EN AW-1070 (Fe < 0.25 wt.%, Si < 0.20 wt.%, Zn < 0.07 wt.%, Mn < 0.03 wt.%, Ti < 0.03 wt.%) was fed simultaneously with a strip of solid austenitic steel type 1.4301 (C 0.07 wt.%, Cr 17.5–19.5 wt.%, Ni 8.0–10.5 wt.%, Mn 2.0 wt.%, Si 0.75 wt.%) into the twin-roll caster, creating 2.5 mm thick strip with Al:steel thickness ratio 4:1. The casting process was realised with a vertical operation plane. No release coatings were applied on the rolls’ surface either before or during the casting process. Temperature of the aluminium melt cladded on the steel substrate was 675 ${}^{\circ }$C and the applied casting rate was set on 4.4 m/min. No preheating was applied on the steel substrate.

The sample for observation of intermetallic layer was annealed in an air furnace at 500 ${}^{\circ }$C for 16 h and subsequently water quenched. Lamellas for transmission electron microscopy (TEM) were prepared by focused ion beam (FIB) in scanning electron microscope Zeiss Auriga Compact. TEM observations were carried out using JEOL 2000 FX and JEOL 2200 FS microscopes. The chemical analysis and elements mapping was accomplished by energy-dispersive spectroscopy (EDS) in TEM.

The phase analysis was performed by evaluation of selected area electron diffraction (SAED) patterns in program JEMS.

The program for diffusion simulation according to Fick’s second law has been developed using freely available FEniCS software [20,21], which is designed for numerical solution of partial differential equations by finite element method. Due to the low D(c) values (order of $10^{-18}$), the diffusion equation is modified to a nondimensionalised form. Concentration function was interpolated by linear Lagrange elements; Crank–Nicolson semi-implicit scheme [22] with Courant–Friedrichs–Lewy condition [23] was applied for time discretisation. The diffusion coefficient D(c) present in the solved diffusion equations was approximated by a piecewise constant function. Since the term D(c) brings nonlinearity to the equation, it was linearised by means of the previous time step. Resolution of 400 nodes was found sufficient for all calculations.

Concentration profiles c(x) obtained by EDS line analysis needed to be smoothed (with Savitzky–Golay filter) for calculation of the diffusion coefficient according to Equation (Equation 2), since c(x) derivative acts as the denominator.

A simplification was introduced into the simulation of diffusion between Al and steel as it was considered as a diffusion in a binary system. The justification for this step is based on comparable values of diffusion coefficients of main elements comprising steel (Fe, Cr, Ni) in Al [24]. In addition, the concentration profiles of these elements resemble each other and the ratio Fe:Cr:Ni remains rather constant through the entire concentration profile.

## 3. Results and Discussion

### 3.1. Transmission Electron Microscopy

After twin-roll casting, no intermetallic layer was observed at the interface between aluminium and steel in our material [25]. Substantial growth of the intermetallic phase occurred after heating at 500 ${}^{\circ }$C [26]. A total 16 h of annealing was chosen for the IMC layer to have sufficient thickness for the phase identification in TEM and, simultaneously, the thickness is not too high for deterioration of mechanical properties (above 10 μm [1]).

Figure 2 represents the TEM micrograph of a FIB lamella prepared from a material annealed at 500 ${}^{\circ }$C for 16 h. Four distinct areas are distinguishable in Figure 2: steel, two phases in the intermetallic layer and aluminium. The interface between steel and IMC is rather flat, whereas the interface between IMC and Al is undulated. This fact would indicate the growth direction from steel towards aluminium.

Figure 3 represents detail of the part of the IMC attached to the steel and its respective SAED pattern. The sub-micrometric grains are slightly elongated in the direction perpendicular to the interface. The polycrystalline diffraction pattern was fitted by ring patterns in the JEMS program as η-phase Al${}_{5}$Fe${}_{2}$ with orthorhombic structure, space group 63 Cmcm and lattice parameters *a* = 0.7652 nm, *b* = 0.6416 nm and *c* = 0.4227 nm [27].

Detail of the IMC phase closer to the aluminium side of the interface is represented in Figure 4. The grains in this part of the IMC layer are columnar, with their long axis perpendicular to the interface of the respective phases. The evaluation of the SAED pattern revealed that it corresponds to monoclinic θ-phase Al${}_{13}$Fe${}_{4}$ with space group 12 C2/m and lattice parameters *a* = 1.5437 nm, *b* = 0.8109 nm and *c* = 1.2430 nm, α = 90${}^{\circ }$, β = 107.66${}^{\circ }$ and γ = 90${}^{\circ }$ [28]. Alternative parameters reported by other authors are *a* = 1.5489 nm, *b* = 0.8031 nm, *c* = 1.2476 nm and β = 107.72${}^{\circ }$ [29] or *a* = 1.550 nm, *b* = 0.808 nm, *c* = 1.250 nm and β = 107.9${}^{\circ }$ [30].

Significant streak structure was observed in the grains of the Al${}_{13}$Fe${}_{4}$ phase (Figure 4a). Tsuchimori et al. [30] described these structures, with the help of high resolution TEM, as planar defects and twins. Three types of planar defects lie on (100), (20$\bar{1}$) and (001) planes and there are three kinds of twin planes—(100), (20$\bar{1}$) and (001). Concerning the (100) planar defect, they were reported to lie on the plane at *x* ≈ 0 or *a*/2. The component of the displacement vector is ≈0 or *a*/2 for the [100] direction and 0.49 nm for the [001] direction. In the case of (100) twin, the reflection is across the plane at *x* ≈ 0 or *a*/2 and the length of the translation is approximately 0.83 nm in the [001] direction. Atomic arrangement around the (20$\bar{1}$) plane is similar to that around the (100) plane, especially for Fe atoms. Nanotwins in the Al${}_{13}$Fe${}_{4}$ phase were also observed by Xu et al. [31].

Furthermore, Tsuchimori et al. [30] showed that many planar defects cause a streak parallel to *c** axis in electron diffraction patterns. This streak penetrates Bragg points that satisfy the conditions that *h* and *k* are odd due to the particular value of the displacement vector [*a*/2,0,0], which is in accordance with our diffractogram (Figure 4b).

### 3.2. Chemical Analysis

Energy dispersive spectrometry mapping in scanning transmission mode was applied in order to describe the relative elemental distribution across the particular phases. Figure 5 shows maps of Al, Fe and Si in Al${}_{5}$Fe${}_{2}$, Al${}_{13}$Fe${}_{4}$ and Al layers and a line scan across the phases. Aluminium dominates all phases; its concentration increases in the direction of the aluminium part of the composite. In the two Al-Fe intermetallic phases, the difference in aluminium content is approximately 3%—it increases from ∼67% in Al${}_{5}$Fe${}_{2}$ to ∼70% in Al${}_{13}$Fe${}_{4}$, reaching almost 100% in the aluminium side of the interface. The increase in Al concentration is compensated by a decrease in iron concentration from ∼23% in Al${}_{5}$Fe${}_{2}$ to ∼20% in Al${}_{13}$Fe${}_{4}$. The other two main elements present in the steel sheet are nickel and chromium. They follow the same trend as iron, reaching concentrations of Ni ∼2%, Cr ∼5%.

Moreover, the distribution map of Si reveals a presence of Si-rich particles decorating the interface between the Al${}_{13}$Fe${}_{4}$ phase and aluminium. These particles are elongated along the interface and form a discontinuous layer. These have also been subjected to SAED analysis to reveal their nature—see Figure 6. They were identified as a body centred cubic structure matching with Al${}_{19}$Fe${}_{4}$MnSi${}_{2}$ structure with space group 204 Im$\bar{3}$ and lattice parameter *a* = 1.256 nm [32]. Manganese in this phase originates from steel; furthermore, Mn can be substituted by Fe or Cr, as these elements are known to substitute for each other in the crystal structures [33].

Si-rich particles at the interface between the Al${}_{13}$Fe${}_{4}$ phase and aluminium can successfully hinder further formation of brittle Al${}_{13}$Fe${}_{4}$ phase [34] and thus help to retain good mechanical properties, which are known to be deteriorated when the IMC thickness at the Al-steel interface exceeds a certain value.

One agglomeration of particles rich in Fe and Si is captured in the aluminium matrix. Such particles are present in Al due to non-negligible impurities content in the alloy and they form during casting.

Xia et al. [35] evaluated the chemical potential of Si in the Fe-Al-Si ternary system in order to clarify the preferential diffusion and aggregation of Si atoms at the Fe/Al interface. They have shown that the chemical potential of Si in the Fe-Al-Si ternary system was lower at the Fe-Al side as compared with aluminium and steel substrate; this fact led to a preferential diffusion of Si to the Fe/Al interface and finally caused the aggregation of Si in the Fe/Al interface.

In order to further evaluate the preferential direction of the growth of the intermetallic phases, the diffusion of Al into steel was numerically simulated.

### 3.3. Diffusion Simulation

The Boltzmann–Matano method was used for the evaluation of the diffusion coefficient D(c) from the concentration profiles of aluminium measured using EDS. The denominator dc/dx acting in Equation (Equation 2) has to be nonzero and monotonous for both physical and numerical reasons. To ensure this and to reduce the noise from measurement, Savitzky–Golay and moving average filters were applied to the measured concentration profile.

Obtained D(c)—Figure 7 should be interpreted as an effective interdiffusion coefficient of aluminium in steel, which could quantify the rate of the ongoing diffusion processes according to Fick’s second law where a concentration gradient acts as the main driving force. The maximum of the diffusion coefficient is reached near the composition of the intermetallic phase layers. This feature enables the intermetallic layer to grow as diffusion proceeds in time.

Similar results were obtained for simulation of the effective interdiffusion coefficient for Fe, calculated from the EDS concentration profile measured in the scanning electron microscope [36].

Figure 8 compares the measured and simulated Al concentration profile. The initial condition (gray line) is represented by a step function. The green line represents the measured concentration profile of aluminium and the red line represents the profile obtained by the numerical simulation. Both measured and computed profiles are in a good agreement. This not only confirms the validity of the obtained interdiffusion coefficient, but it also enables one to see which direction the interface moves during the annealing. Measurement of the concentration profile itself after a certain time of annealing does not carry the information about the initial position of the interface between steel and Al. However, the ability of diffusion coefficient determination and subsequent solving of the diffusion equation with this D(c) could show it when the initial condition c(t=0,x) is compared to the final solution c(t,x).

Since the published value of iron diffusivity in aluminium is higher than that of aluminium in iron [24], researchers are often inclined to a conclusion that diffusion proceeds towards aluminium and related formation of a layer comprised of intermetallic phases grows in Al [24,37,38]. The plot of the simulated concentration profile together with the initial condition also shows the pronounced growth of intermetallic layer towards the Al side, although not fully unambiguously—the simulation shows the growth on both sides of the Al-steel interface, but it is less developed on the steel side (see the plateau region in the concentration profile displayed in Figure 8).

An uncertainty is brought by the fact that an intermetallic phase forms as the diffusion proceeds, a question arising as to whether the used description of diffusion is sufficient. Firstly, the Boltzmann–Matano relation is based on the assumption that the concentration profile in *t* = 0 s fits the step function [17,39]. However, the initial concentration profile after the twin-roll casting process probably evinces some consequences of the first rapid diffusion processes ongoing in the first stages of the clad sheet production when the Al melt was in a direct contact with the steel substrate. Secondly, the question of the applicability of the Boltzmann–Matano method still persists, since the condition of the proportionality of gradients of chemical potential and concentration is presented to be fulfilled only in dilute systems and ideal solid solutions. In the case of the Al-steel joint, a layer of several intermetallic phases forms during diffusion, which makes the system more complicated.

## 4. Conclusions

Aluminium-steel clad material was subjected to annealing at 500 ${}^{\circ }$C. The intermetallic layer, which formed on the aluminium-steel interface, was investigated by transmission electron microscopy. Three different phases were identified:Orthorhombic phase Al${}_{5}$Fe${}_{2}$ at the steel side of the interface;Monoclinic phase Al${}_{13}$Fe${}_{4}$ with columnar grains and high density of planar faults and twins;Cubic phase isostructural with Al${}_{19}$Fe${}_{4}$MnSi${}_{2}$ decorating interface between Al${}_{13}$Fe${}_{4}$ and aluminium in a discontinuous layer.

The effective interdiffusion coefficient for aluminium was calculated using the Boltzmann–Matano method from the shape of the concentration-depth profile. Its maximum is reached near the composition of the intermetallic phase layers. Finite element method simulation of diffusion with the calculated interdiffusion coefficient shows that the intermetallics grow rather towards the aluminium side of the interface.

## Figures and Tables

**Figure 1 materials-14-07771-f001:**
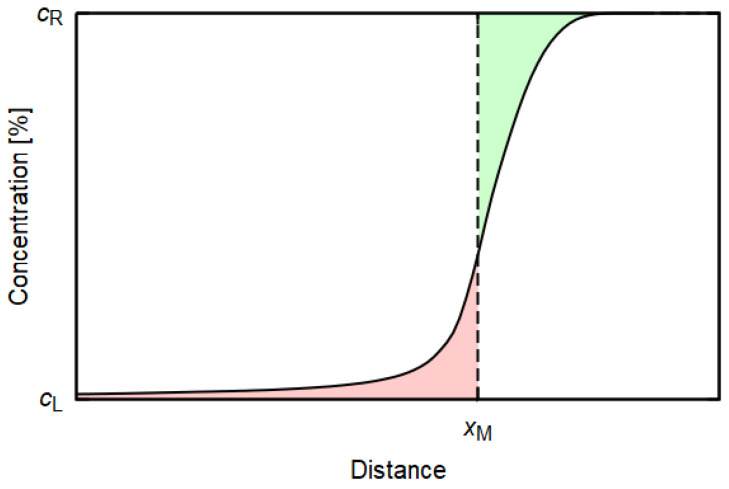
Diagram showing the position $x_{M}$ of the Matano plane. The green and red areas are equal to each other.

**Figure 2 materials-14-07771-f002:**
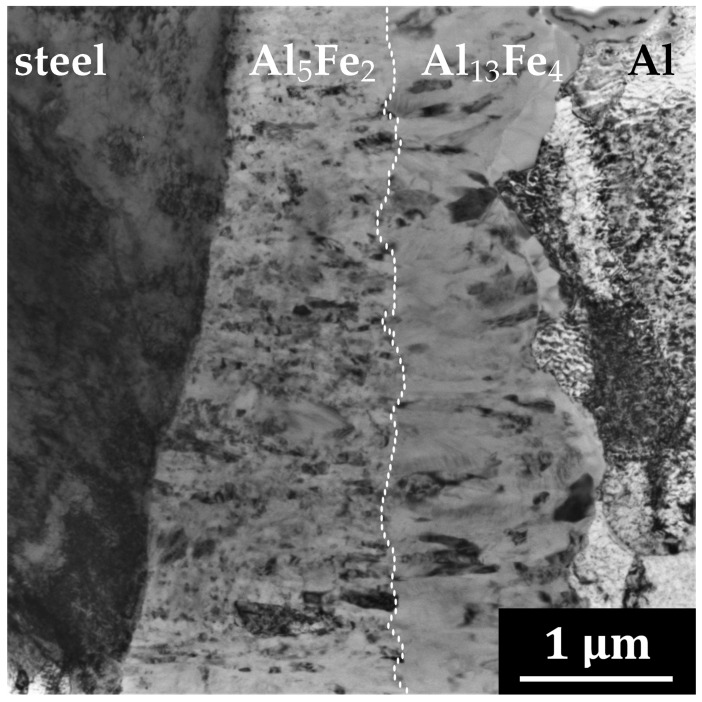
TEM image of the Al-steel interface after 16 h annealing at 500 ${}^{\circ }$C. Identified phases from left: steel, Al${}_{5}$Fe${}_{2}$, Al${}_{13}$Fe${}_{4}$ and Al.

**Figure 3 materials-14-07771-f003:**
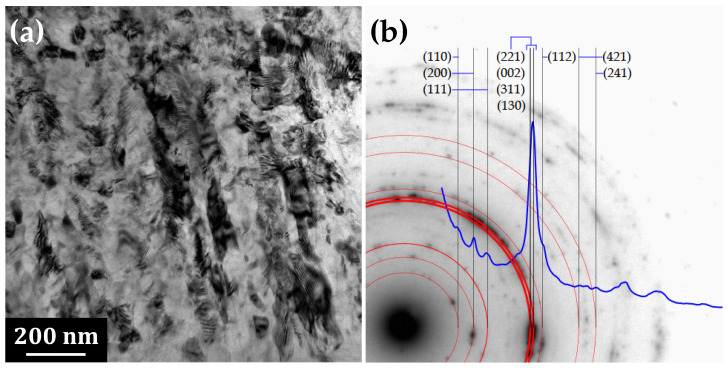
(**a**) TEM image of the intermetallic phase Al${}_{5}$Fe${}_{2}$, fine-grained layer closer to the steel side of the interface. (**b**) Ring diffraction pattern: blue line shows over-circles-integrated intensity of the experimental diffractogram, vertical lines represent the reciprocal interplanar spacing of Al${}_{5}$Fe${}_{2}$ phase (symmetry space group Cmcm) corresponding to selected (hkl)-s.

**Figure 4 materials-14-07771-f004:**
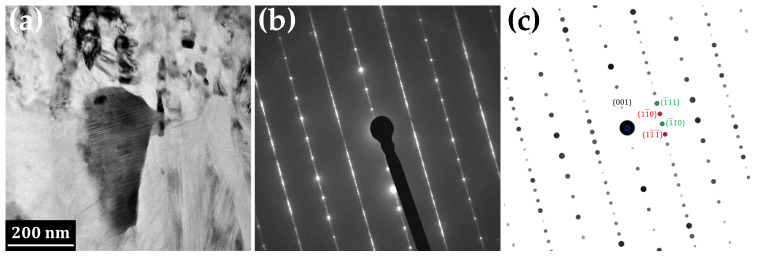
(**a**) TEM image of the columnar intermetallic phase Al${}_{13}$Fe${}_{4}$ (symmetry space group C2/m) adjacent to aluminium, (**b**) SAED, (**c**) simulation of SAED of two twins with twin plane (001): green hkl indices belong to the grain in orientation [110], red ones to the second grain in orientation [$\bar{1}\bar{1}$0]. The streaks in (**b**) are caused by planar defects.

**Figure 5 materials-14-07771-f005:**
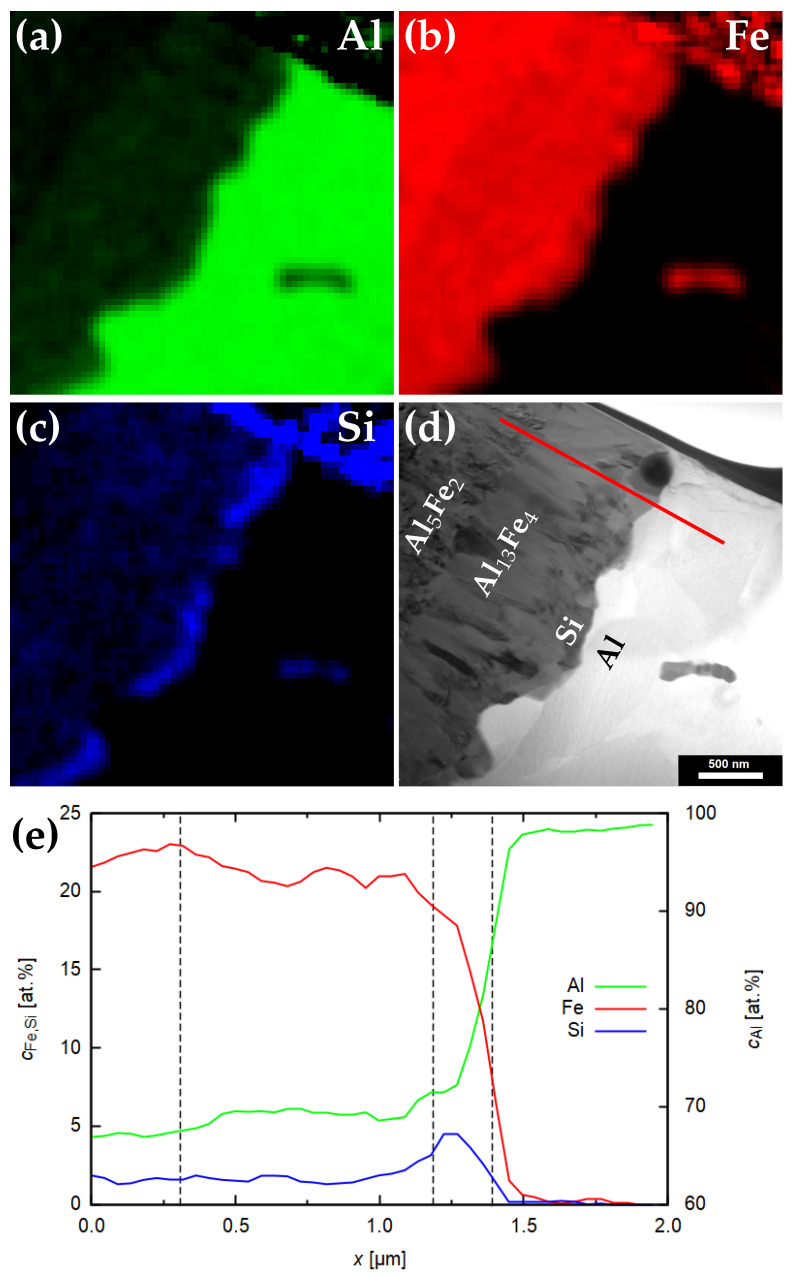
Mapping of chemical elements by EDS across the IMC phases. (**a**) Al, (**b**) Fe, (**c**) Si, (**d**) respective STEM image (Si denotes the Al${}_{19}$Fe${}_{4}$MnSi${}_{2}$), (**e**) concentration profile along the red line marked in (**d**). Vertical lines denote interfaces between respective phases.

**Figure 6 materials-14-07771-f006:**
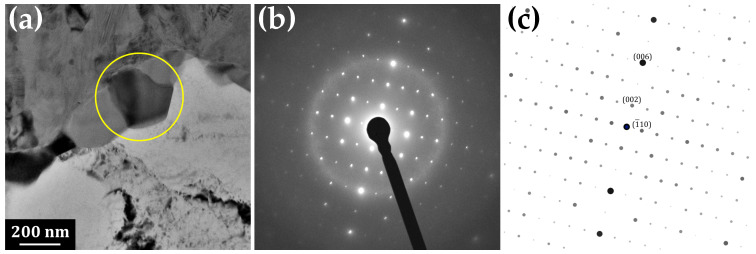
(**a**) TEM image of the Si-rich particle between Al${}_{13}$Fe${}_{4}$ and Al. (**b**) SAED. (**c**) Fitting of the SAED with α-phase Al${}_{19}$Fe${}_{4}$MnSi${}_{2}$ (symmetry space group Im$\bar{3}$), zone [110].

**Figure 7 materials-14-07771-f007:**
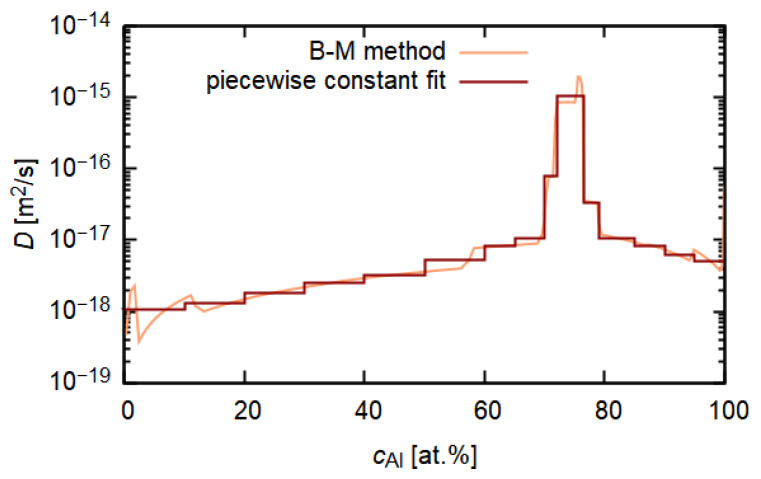
Concentration dependence of diffusion coefficient of aluminium in steel obtained from a measured concentration profile after 16 h of annealing at 500 ${}^{\circ }$C simulated by Boltzmann–Matano (B-M) method.

**Figure 8 materials-14-07771-f008:**
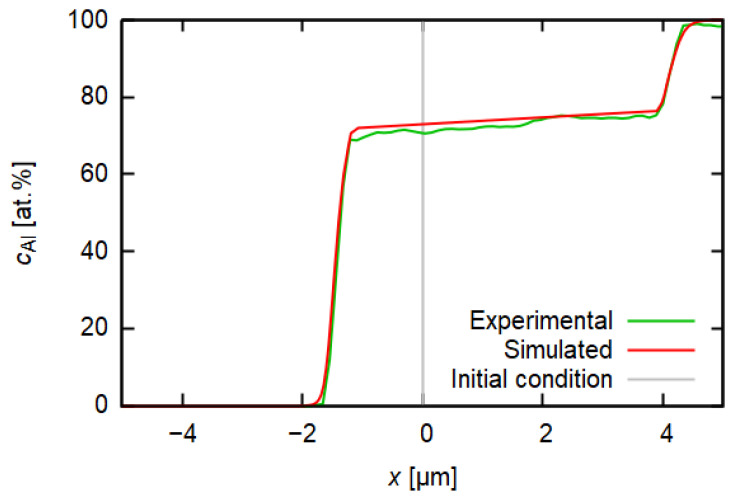
Measured (EDS) and simulated (finite element method) concentration profiles for aluminium after 16 h of annealing at 500 ${}^{\circ }$C.

## Data Availability

The data underlying this article will be shared on request from the corresponding author.

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
