# Peer review of "Intermetallic Phases Identification and Diffusion Simulation in Twin-Roll Cast Al-Fe Clad Sheet"

_materials, 2021, doi:10.3390/ma14247771_

Round 1

Reviewer 1 Report

The paper is very well written. In terms of presentation, only very minor changes are required. "Al:Fe ratio 4:1" is mentioned. Is this thickness ratio or atomic ratio? The materials used were pure aluminium EN AW-1070 and austenitic steel type 1.4301, but not many people are familiar with the material designations and it would be good to give the chemical compositions in the manuscript. Temperatures of the Al melt and solid austenitic steel should be clearly specified.

The paper focuses on characterization of intermetallic compounds between Al and an austenitic steel. The characterization is of value; however, it is a pity that it is not discussed at all how the processing parameters would affect the formation of intermetallic compounds and what kind of intermetallic compounds would be desirable for the bonding of the dissimilar materials. Adding such discussion could improve the paper considerably. 

Author Response

Dear reviewer,
thank you for your valuable comments. 
The section "2. Materials and Methods" was modified according to your comments. The composition of both materials is given. The phrasing was modified to "Al:steel thickness ratio 4:1". Temperatures of Al melt (675 °C) and steel substrate are given (room temp.).
Paragraph discussing the influence of the casting parameters (steel substrate temperature) and ideal IMC thickness is added to Introduction.

Reviewer 2 Report

The paper investigate that intermetallic phases identification and diffusion simulation in twin-roll cast Al-Fe clad sheet. I think that the authors need to present more scientific explanations on the mechanism of Al diffusion.

  1. Why did the author select the annealing temperature of 500 oC ? The author should give reasonable reason.
  2. In Figure 2, the TEM figure needs to identify the interface for Al5Fe2 and Al13Fe4 by SAED.
  3. The author should give scientific explanation for the Si. Where did the Si diffuse?
  4. In Figure 5 and 6, it didn't observe any identification for the Mn. The author also needs to explanation. 

Author Response

Dear reviewer,
thank you for your valuable comments. 

1) The annealing temperature and time were choosen according to our previous research cited in the paper. We have choosen 500 °C and 16 hours as the thickness of the IMC layer is large enought to clearly distinguish between respective phases in TEM and not too large to deteriorate mechanical properties of the composite (above 10 um). This explanation was added to the beginning of section 3.1.

2) The interface between Al5Fe2 and Al13Fe4 was identified and marked by a white line in Figure 2.

3) The direction of diffusion of Si was explained in the terms of chemical potential and respective citation was provided (end of Section 3.2).

4) I agree that it is confusing, that no Mn is mentioned in the analysis, when it is presented in the identified phase. I have reformulated the statements, that the observed phase is isostructural with the Al19Fe4MnSi2 phase. Iron and chromium (~18 wt.% in the used steel) are known to substitute for Mn in the crystal structures - the citation is provided in the text.

Round 2

Reviewer 2 Report

All comments have modified and replied. The paper could be accepted as this revised form.